# Current Techniques for Fruit Juice and Wine Adulterant Detection and Authentication

Hoa Xuan Mac [1,2], Thanh Tung Pham [1,3], Nga Thi Thanh Ha [1,2], Lien Le Phuong Nguyen [1,4], László Baranyai [1,*] and László Friedrich [1]

1 Institute of Food Science and Technology, Hungarian University of Agriculture and Life Sciences, 1118 Budapest, Hungary; hoamx@hufi.edu.vn (H.X.M.); tungpt@hcmute.edu.vn (T.T.P.); ngahtt@hufi.edu.vn (N.T.T.H.); nguyen.le.phuong.lien@uni-mate.hu (L.L.P.N.); friedrich.laszlo.ferenc@uni-mate.hu (L.F.)
2 Faculty of Food Science and Technology, Ho Chi Minh City University of Industry and Trade, Ho Chi Minh 700000, Vietnam
3 Faculty of Chemical and Food Technology, Ho Chi Minh City University of Technology and Education, Ho Chi Minh 700000, Vietnam
4 Industrial University of Ho Chi Minh City, Ho Chi Minh 700000, Vietnam
* Correspondence: baranyai.laszlo@uni-mate.hu

**Abstract:** Fruit juice and wine are important beverages that are consumed all over the world. Due to their constantly increasing demand and high value, fruit juice and wine are one of the most frequent targets of adulteration. Since adulterated foods are proven to have harmful effects on health, several approaches have been utilized for the detection of fruit juice and wine adulteration. Based on the requirement for sample destruction, analytical techniques to assess food authenticity can be classified into 2 main categories, i.e., destructive and non-destructive techniques. This paper provides an overview on the principle of adulteration detection, its application and performance, and the advantages and limitations of various analytical techniques. Destructive approaches, such as physicochemical methods, isotope analysis, elemental analysis, chromatographic techniques, and DNA-based techniques, are reviewed. Furthermore, non-destructive approaches, including spectroscopic-based techniques, nuclear magnetic resonance spectroscopic technique, electronic techniques, and imaging-based techniques, are discussed.

**Keywords:** food adulteration; food control; food fraud

## 1. Introduction

According to the Codex Alimentarius Commission, fruit juice is defined as "the unfermented but fermentable liquid obtained from the edible part of sound, appropriately mature and fresh fruit or of fruit maintained in sound condition by suitable means including postharvest surface treatments applied in accordance with the applicable provisions of the Codex General Standard" [1]. Fruit juices are consumed worldwide and have become very popular due to their nutritional value and variety of beneficial health effects. By 2023, the amount of juice consumption is expected to be 36,809.8 million liters worldwide [2]. In addition to macronutrients and micronutrients, fruit juices contain active compounds that can boost immunity and provide a range of other health benefits [3]. Fruit juice consumption is proven to have effects in preventing the development of a wide range of diseases, such as cardiovascular disease, cancer, and neurodegenerative diseases [4]. To protect the consumers from purchasing inferior products with misleading description, the quality and the safety of juice products are consistently regulated by comprehensive legislation to ensure that all necessary information regarding their nutritional benefits and compositions are provided. In Europe, the standards for juice products, including their quality, composition, production, and labeling, are governed by a specific European

Fruit Juice Directive (Directive 2012/12/EU). Furthermore, the Reference Guidelines of the European Fruit Juice Association (AIJN) Code of Practices with regard to quality, authenticity, and identity have been established [5].

Wine is a type of alcoholic beverage, which is produced through the process of partial or complete alcoholic fermentation of fresh grapes or grape must [6]. Wine is an important beverage in worldwide trade with the estimated global wine production and consumption of 262 and 235 million hectoliters in 2021, respectively [7]. Moderate consumption of wine can offer various beneficial health effects to consumers [8]. The value of wine is affected by many factors including geographical origin, grape variety, vintage, and production methods [9]. In order to protect consumers and to evaluate the wine quality, the European Commission has introduced a regulatory framework for wines and spirits and quality schemes for food products (Council Regulation, EEC No. 510/2006), linking products to their geographical origins [10]. In addition, International standard for the labelling of wines has been launched by International Vine and Wine Organization (OIV), to ease international exchange and to ensure fair information to consumers [11].

Due to their high economic values and trade volumes, fruit juices and wines have become one of the most frequent targets for adulteration. The term "adulteration" in beverage is generally used to describe different types of fraud, including dilution with water, addition of exogenous substances (such as sugars, alcohol, organic acids, and coloring and flavoring agents), substitution with lesser quality products, and mislabeling in relation to variety and origin [12]. Negative effects of adulterated foods on human health was reviewed by Bansal et al. (2017) [13]. Besides health hazards, food adulteration also results in economic costs, and the lost sales of food businesses were estimated at 2–15% of annual revenues [14].

From farm to table, various methods have been used to fight the food fraud. The adoption of analytical methods has emerged as the most effective way to prevent adulteration. Based on the requirement for sample preparation, analytical techniques used to assess food authenticity can be classified into two main categories, including destructive and non-destructive techniques. This article is focused on reviewing the use of current techniques in detection of fruit juice and wine adulteration, following these main categories.

## 2. Destructive Techniques

### 2.1. Physicochemical Methods

For detection of juice fraud, physicochemical analysis is frequently used by the quality control staff in production plants, and quality parameters are assessed, such as total soluble solids (TSS), titratable acidity (TA), pH, organic acid content, and mineral content. Adulteration by water dilution and sugar addition can be detectable through measurement of TSS. More accurate detection of sugar addition can be obtained using a reducing sugar test, based on the fact that reducing sugars often account for 80 to 90% of TSS. Additionally, fruit juices adulterated with pulp wash usually have lower concentration of primary amines than authentic juice. Hence, the formol or formaldehyde test can be used to distinguish 100% juices and adulterated ones [15]. In case of wine, some physicochemical methods can be used for disclosing adulteration, including measurement of TSS using hydrometry and/or refractometry, quantification of ethanol with hydrometry, determination of organic acid concentration by measuring TA, determination of volatile acidity using steam distillation and titration, and determination of reducing sugars using reducing sugars test with copper [16].

Despite its low cost and broad application in quality control, the physicochemical methods exhibit limited detection of low-level adulteration. Wang et al. (2016) [17] prepared adulterated lemon juices by adding a solution containing citric acid (5%, $w/w$) and sucrose (6%, $w/w$). Total TA, °Brix value, and pH values were measured to distinguish between the lemon juice samples. The results indicated that these traditional quality indicators of juice were not able to distinguish the pure juice and the adulterated ones [17]. In another study by Vitalis et al. (2020) [18], TSS was measured using a digital pocket refractometer to

identify adulterants in tomato concentrate at relatively low concentration (0.5–10%). Their finding also revealed that the used method was not suitable for detection of adulteration below a certain level [18].

Overall, physiochemical methods are simple in measurement and appropriate for initial monitoring of adulteration in production plants. However, they are not good choices for detection of sophisticated types of fraud and low adulteration levels.

### 2.2. Isotope Analysis

Geographical origin, variety, methods of cultivation, and production methods create the unique identities of isotope ratios in authentic juice and wine. Thus, isotopic methods based on the analysis of isotope ratios can be used for juice and wine authentication. Table 1 summarizes typical examples of isotopic methods of detection for authentication and for adulteration of juice and wine.

**Table 1.** Typical application of isotopic and elemental techniques for authentication of and adulteration detection in juice and wine.

| Sample | Technique | Aim | Accuracy | Reference |
|---|---|---|---|---|
| Lemon juices | Isotope analysis | Authenticity of juices | - | [19] |
| Lemon juice | Isotope analysis | Adulteration detection | Detection limit: 10% natural citric acid was replaced with exogenous citric acid | [20] |
| Fruit and vegetable juice | Isotope analysis | Detection of sugar and water addition | Detection limit: 20% for water and 7% for sugar | [21] |
| Wine | Isotope analysis | Detection of sugar and water addition | Classification correction: 100% | [22] |
| Wine | Isotope analysis | Authenticity of wines | Classification correction: 93.1% (ANN) and 83.9% (DA) | [23] |
| Wine | Isotope analysis | Authenticity of wines | Classification correction: 98.2% | [24] |
| Apple and orange juice | Elemental and isotope analysis | Discrimination of juices | Classification correction: 93.3% for apple juice and 90% for orange juice | [25] |
| Orange juice | Elemental analysis | Authenticity of juices | - | [26] |
| Wine | Elemental analysis | Authenticity of wines | PCA classification: 83% using first three principal components (PCs) | [27] |
| Wine | Elemental analysis | Authenticity of wines | Classification correction: 96% | [28] |
| Wine | Elemental analysis | Wine authentication | Classification correction: 94% | [29] |

Isotope analysis is widely applied to detect juice adulteration with sugars, organic acids, and water. For instance, to detect adulteration in lemon and lime juices, Guyon et al. (2014) [19] optimized an analytical protocol to determine $\delta^{13}C$ values of organic acids and sugars in 35 samples collected from different geographical origins using high performance liquid chromatography linked to isotope ratio mass spectrometry (HPLC-IRMS). The average $\delta^{13}C$ values of a mixture including citric acid, glucose, and fructose were found to be $-25.40 \pm 1.62‰$, $-23.83 \pm 1.82‰$ and $-25.67 \pm 1.72‰$, respectively. These ranges of $\delta^{13}C$ values were then used to verify the adulteration in commercial lemon and lime juices. The results revealed that 10 in 30 commercial juice samples contained added organic acids or sugars as they had $\delta^{13}C$ values outside the reference ranges [19]. Additionally, in a study by Bononi et al. [20], the $\delta^{13}C$ values for organic acids and sugars in 20 genuine lemon juices derived from two regions of Italy were determined using HPLC-IRMS. After measuring these genuine samples, four isotopes were used to identify the natural range of these components in Italian lemon juice. The exogenous addition of these compounds to commercial lemon juice (42 samples) was investigated using these isotopes. Finally, the adulteration of lemon juices was detected due to more positive values of $\delta^{13}C$ resulting

from the addition of citric acid and sugar [20]. In a recent study by Wu et al. [21], $\delta^{13}$C and $\delta^{18}$O values of 21 fruit and vegetable juices were analyzed to detect the addition of water and sugar to not-from-concentrate juice. The authors confirmed that their method was able to determine more than 20% added water and more than 7% extraneous sugar in juice samples [21]. Nevertheless, the generalization of their results was limited because the number of studied samples was quite small.

In detection of wine adulteration, the site-specific natural isotopic fractionation with $^2$H nuclear magnetic resonance (SNIF-NMR) method based on isotopic ratio of deuterium/hydrogen (D/H) and with IRMS based on determination of $^{13}$C/$^{12}$C ratio of ethanol and $^{18}$O/$^{16}$O ratio of water have become the official methods for assessment of wine fraud in the European Union (EU) [30]. This method has been applied in wine authentication by various authors. For example, Geana et al. [22] initially investigated 23 authentic wines and found their $\delta^{13}$C values in the range of −29.19‰ to −25.19‰ and their $\delta^{18}$O values in the range of 0.71‰ and 4.38‰. Then, the investigation was continued on 29 commercial wines, and their authenticity was verified using the identified range of authentic wines. Among the commercial products, 16 samples were identified as ''good wine'', and the remaining samples were identified as "adulterated wine" and ''suspect wine'' [22]. As obtained data from ''suspect wine'' was quite close to natural range of authentic wines, more information on weather conditions at ripening and harvesting time is required to draw an accurate conclusion. Using the same analytical method but by combining with chemometrics, Wu et al. [23] determined the $^{13}$C/$^{12}$C ratio of ethanol and glycerol and the $^{18}$O/$^{16}$O ratio of water in wine to discriminate 600 imported wine samples in China. Three multivariate methods, including artificial neural network (ANN), discrimination analysis (DA), and random forest (RF), were used to develop classifiers. The results showed that ANN outperformed the two remaining methods with an accuracy of up to 93.1%; RF was found unsuitable for wine origin traceability in their study [23]. In another study by Wu et al. [24], the $\delta^{13}$C values of wine ethanol and glycerol and the $\delta^{18}$O values of the wine water were analyzed in an attempt to develop a classification tool for the verification of geographic origin of 240 French red wines, using machine learning models (ANN and DA). The results showed that only ANN method with an accuracy of 98.2% was suitable for differentiating red wines [24]. Obviously, using chemometrics provided researchers a better understanding about the discrimination performance of the applied method.

Although isotope analysis has high sensitivity, high accuracy, and low detection limit, its accuracy can be reduced due to the instability of isotope ratios during processing and storage of products. Additionally, high equipment price is another limitation of this technique.

*2.3. Elemental Analysis*

Natural juice is characterized by a certain range of mineral levels. Therefore, elemental analysis can be also applied for detection of juice adulteration based on element markers [31]. Table 1 summarizes some typical examples of the use of elemental methods for detecting adulteration of juice and wine. Cristea et al. [25] stated that the high potassium content in juice can relate to the addition of sweeteners (like acesulfame K) or preservatives (like potassium benzoate and potassium sorbate); meanwhile, low potassium content can be an indication of water dilution. Additionally, high calcium content in orange juice can be the result of pulp addition as calcium concentration in pulp is higher than in juice [25]. In a study by Schmutzer et al. [26], elemental profiles of 23 commercial orange juices were analyzed to evaluate their authenticity using inductively coupled plasma mass spectrometry (ICP-MS). Though some of the juice samples were labeled as "100% fruit juice", the results revealed that all juices had a ratio of K to Mg of less than 50; it meant that they were adulterated with exogenous sugar [26]. In addition, a combined data of elemental and isotope analysis was used by Cristea et al. [25] who adopted ICP-MS to discriminate commercial and freshly squeezed apple and orange juices. The supervised classification method of linear discriminant analysis (LDA) was applied to evaluate the differences between juice

samples. A satisfactory classification accuracy above 90% (validation) was obtained for both apple and orange juices. The contents of K and Na were the most important variables for discrimination of apple juice, while Na content provided the most contribution to the result. In all cases, the isotopic ratio of oxygen was the most significant variable [25].

The elements in wine are derived from endogenous (grape variety and maturity and climatic conditions) and exogenous sources (external impurities from different winemaking procedures); therefore, the wine authentication can be performed based on the analysis of minerals. The effectiveness of elemental analysis was confirmed by Geana et al. [27] who used ICP-MS for the analysis of elemental composition to differentiate 60 wine samples from three main wine production regions of Romania. They pointed out that the contents of five analyzed elements, including Mn, Cr, Sr, Ag, and Co, were the most useful for differentiating wines. The principal component analysis (PCA) model with three first PCs could separate the wine samples using 83% of the total variance of the acquired data [27]. Their classification result could be more accurate, if supervised models were applied. In addition, a combination of ICP-MS with multivariate statistical analysis (PCA and LDA) was used by Azcarate et al. [28] for differentiation of wine from different Argentinean regions. Based on the evaluated elemental profile (Ba, As, Pb, Mo, and Co), the proposed method allowed correct discrimination in terms of the geographical regions. Accordingly, the PCA result explained 95.95% of the variance of the total obtained data; and LDA model reached an accuracy higher than 96% [28]. Nevertheless, their findings would be strengthened when the correlation between the element profile of soil and wine was considered. In another study, inductively coupled plasma optical emission spectrometer (ICP-OES) was adopted by Rodrigues et al. [29] to distinguish 111 sparkling wines from four countries (Brazil, Argentina, France, and Spain), based on elemental profile (Al, B, Ba, Ca, Cu, Fe, K, Li, Mg, Mn, Na, and Sr). A result of 94% accurate classification was achieved by the authors, using the three key elements of B, K, and Na [29].

Like the isotopic method, this technique has the advantages of high accuracy and low detection limit. However, the high costs of sample pre-treatment and the high requirement for the experimental operation are the disadvantages of elemental analysis.

### 2.4. Chromatographic Techniques

Chromatography is a reliable analytical approach that is suitable for the identification of adulterants in food. By utilization of chromatographic techniques, a targeted sample containing a mixture of various compounds is separated and detected. For quality control of beverages, the most used detection methods are flame ionization detector (FID) and mass spectrometry (MS) [32]. The validation of authentication of products can be performed through determination of specific marker compounds or fingerprinting analysis. The sensitivity and high separation efficiency of chromatography in authentication have been proven through many studies; however, the technique faces some limitations such as complex procedure for sample pretreatment and operation and possible loss of instable compound [33]. Some applications of gas and liquid chromatography on fraud detection of juice and wine are presented in Table 2.

#### 2.4.1. Gas Chromatography (GC)

Numerous studies on the application of gas chromatography (GC) for detection of fruit juice adulteration have been reported so far. For example, Yamamoto et al. [34] used γ-terpinene and linalool as chemical markers for detecting the addition of Shiikuwasha juice to calamondin juice with gas chromatography–mass spectrometry (GC-MS), and the lowest detection level of 1% was reported. Willems and Low [35] developed a method using capillary GC with FID for detecting the addition of pear juice in apple juice using oligosaccharide and arbutin as marker. Consequently, a low adulteration level of 0.5–3% was achieved by the proposed method [35]. In another study by Nuncio-Jáuregui et al. [36] on the identification of pomegranate juice adulteration with peach and grape juice, the method of headspace solid phase microextraction (HS-SPME) was optimized to extract the

volatile compounds. GC-MS was then used to isolate and identify volatile profile. Based on the variation of specific volatile compounds (acetic acid, isoamyl butyrate, 1-hexanol, and linalool for added grape juice samples; butyl acetate, isobutyl butyrate, benzyl acetate, and isoamyl butyrate for added peach juice sample), the authentication of pomegranate juice was achieved with the lowest detectable levels of 10% for peach juice and 50% for grape juice [36]. Recently, the authenticity of premium organic orange juices was confirmed by Cuevas et al. [37] using fingerprinting analysis (HS-SPME coupled with GC-MS) combined with chemometric methods. As a result, the mid-level data fusion partial least squares-discriminant analysis (PLS-DA) model was found appropriate for authentication with the accuracy of 100% [37].

**Table 2.** Typical application of chromatographic techniques for authentication of and adulteration detection in juice and wine.

| Sample | Technique | Aim | Accuracy | Reference |
|---|---|---|---|---|
| Shiikuwasha juice | GC | Detection of juice-to-juice adulteration | Detection limit: 10% | [34] |
| Apple juice | GC | Detection of pear juice addition to apple juice. | Detection limit: 0.5–3% | [35] |
| Pomegranate juice | GC | Detection of juice-to-juice adulteration | Detection limit: 10% for added peach juice and 50% added grape juice | [36] |
| Orange juices | GC | Juice authentication | Classification correction: 100% | [37] |
| Wine | GC | Discrimination of wines | Discrimination rate: 100% | [38] |
| Wine | GC | Wine authentication | Detection limit: 0.03–10.03 g/L | [39] |
| Wine | GC | Detection of adulteration | Detection limit: 0.1–2 mg/L | [40] |
| Citrus fruit | LC | Differentiation of juices | Classification correction: 100% | [41] |
| Purple grape juice | LC | Detection of added apple juice in purple grape juice | - | [42] |
| Wine | LC | Discrimination of wines | Classification correction: 88% of the total variance using two first PCs in PCA. | [43] |
| Wine | LC | Authentication of wines | Classification correction: 100% | [22] |
| Wine | LC | Wine authentication | Classification correction: 95.4% | [44] |
| Wine | LC | Authentication of wine | Classification correction: >90% | [45] |

　　　When it comes to wine, this type of beverage contains a variety of compounds that originate from grapes, the alcoholic fermentation, and the aging of wine. Several of these compounds have effects on wine aroma and can be used as a "fingerprint" to authenticate wines. Many studies on GC technique have been conducted, focusing on differentiating wines according to geographic origin and grape variety, and detecting wine adulteration. For example, Welke et al. [38] discriminated five types of wines with different grapes based on the volatiles obtained using a combination of HS-SPME and comprehensive two-dimensional gas chromatography with time-of-flight mass spectrometry detection (GC×GC/TOFMS). This two-dimensional GC system allowed the authors to obtain better separation capabilities compared to one-dimensional GC. Twelve extracted volatile compounds were useful to distinguish the wine samples with accuracy of 100% using LDA model [38]. In another study, Langen et al. [39] utilized a heart-cut multidimensional GC-MS system to determine α-ionone, β-ionone, and β-damascenone in various authentic and commercial wines. Their finding revealed that an elevated concentration of these compounds in wine samples can be served as indicators of adulteration (suggested thresholds: content of α-ionone > 0.003 μg/L, content of β-ionone > 1 μg/L, and content of β-damascenone > 10 μg/L). Moreover, the enantiomeric ratio of α-ionone could be used as an adulteration marker because the addition of exogenous α-ionone resulted in the change

of this ratio [39]. Sagandykova et al. [40] developed a method using SPME-GC–MS for the detection of semi-volatile additives (propylene glycol and sorbic and benzoic acids) in commercial wines by optimization of SPME method. The optimized approach showed good performance in terms of linearity (coefficient of determination, $R^2 > 0.98$) when a method of standard addition was performed. The linear ranges for the detection of propylene glycol and sorbic and benzoic acids were 0–250 mg/L, 0–125 mg/L, and 0–250 mg/L, respectively. Using the developed method, three in twenty-five wine samples were found adulterated with propylene glycol and sorbic and benzoic acids [40].

### 2.4.2. Liquid Chromatography (LC)

According to the literature, the application of LC on assessing the addition of adulterants to fruit juice can be accomplished based on the analysis of juices' phenolic profile and anthocyanin profile. For example, Abad-García et al. [41] analyzed polyphenols profile of citrus juices (orange, tangerine, lemon, and grapefruit juices) for the assessment of authenticity, using a reversed-phase high performance liquid chromatography (HPLC) with photodiode array detection. The acquired polyphenolic profiles were then used to develop LDA and PLS-DA classification models. As a result, the LDA model obtained 100% accuracy using four best variables (naringenin-7-*O*-rutinoside-4′-*O*-glucoside, naringenin-7-*O*-rutinoside, hesperetin-7-*O*-rutinoside, and apigenin-6,8-di-*C*-glucoside). For PLS-DA model, 100% accuracy was also reported with the most significant variables of naringenin-7-*O*-rutinoside-4′-*O*-glucoside, apigenin-6,8-di-*C*-glucoside, isosakuranetin-7-*O*-rutinoside, and naringenin-7-*O*-rutinoside [41]. Phloridzin, a phenolic compound, is not naturally available in grapes; while this compound is present in apples in a higher proportion compared to other fruits. Based on this fact, Spinelli et al. [42] used HPLC with a photodiode array detector to analyze phloridzin for detecting the addition of apple juice to purple grape juice. Their results indicated that the proposed method allowed the detection of apple juices in adulterated grape juices with phloridzin content of 4.54 to 8.39 mg/L [42].

Like in juice, phenolic compounds are also found in grapes as well as in must and wine. Since the compositions of polyphenolic compounds in wine vary widely, depending on the grape varieties, the winemaking process, and the climatic conditions; they can be used as markers for authenticating wine [46]. For instance, HPLC method based on analysis of anthocyanin profile was utilized to discriminate different wines (Brazilian tropical wines, Brazilian temperate wines, and temperate Chilean wines) by de Andrade et al. [43]. As a result, the concentrations of nine anthocyanins were determined and used as the classification factor for PCA. The PCA results showed that the two first PCs accounted for 88% of the total variance. The contents of petunidin-3-glucoide, peonidin-3-glucoside, and malvidin-3-glucoside contributed the most to PC1; while the contents of peonidin-3-glucoside coumarate, peonidin-3-glucoside-acetate, and malvidin-3-glucoside primarily represented PC2 [43]. In another study, Pavloušek et al. [44] succeeded in classifying 43 different wines with a HPLC method for analyzing non-flavonoid phenolic compounds. Results indicated that the canonical discriminant analysis allowed them to classify the wine with 95.4% correction rate [44]. Similarly, Geana et al. [22] applied a HPLC system to characterize the anthocyanin profile for authenticating wine. An LDA model was developed to differentiate the wine samples with respect to concentrations of anthocyanins and defined anthocyanins ratios. Differentiation results of LDA achieved 100% accuracy using two discriminant factors. The most significant anthocyanins for discrimination included individual anthocyanins of delphinidin-3-*O*-glucoside, petunidin-3-*O*-glucoside, peonidin-3-*O*-glucoside, malvidin-3-*O*-glucoside, and peonidin-3-*O*-(6-p-coumaroyl) glucoside [22]. Recently, Zhi et al. [45] developed an analytical method that combined three-way HPLC with diode array detection and chemometrics (PCA-LDA) to distinguish wines according to their vintages. The proposed method could prevent the loss of analytes and thus increase the accuracy of analysis. As a result, an discrimination rate above 90% was achieved [45].

The differences in polyphenolic profile of juices are sometimes not caused by adulteration but by climatic conditions, the environment, the processing technology, and the

degree of fruit ripeness. Furthermore, the oxygen stability of anthocyanins and betacyanins can significantly change due to the activities of native polyphenol oxidases when these compounds are used as markers [31]. Therefore, these factors need to be taken into account in the analysis of polyphenols profile.

### 2.5. DNA-Based Techniques

Since DNAs of food products maintain their stability under conditions of environment and cultivation and production process, DNA-based methods have become reliable means for food authentication [47]. DNA polymorphisms resulting from natural mutations of genetic code can be used to identify plant species. In DNA-based techniques, the DNA fragment of interest is extracted from objective samples, and then the specific genetic polymorphisms are amplified to obtain the amplicons, followed by an analysis of the obtained amplicons to reveal the characteristics of polymorphisms [48]. For authentication and detection of food adulteration, the most commonly used DNA-based methods are polymerase chain reaction (PCR), real-time PCR, high resolution melting (HRM) analysis, microarrays, and next generation sequencing [49]. Table 3 presents the applications of DNA-based techniques for authenticating and detecting adulteration in juice and wine.

In the study of juice adulteration, orange juices have gained the most interest from researchers due to their market value. Most of the studies focused on disclosure of juice-to-juice adulteration. For instance, a PCR restriction fragment length polymorphism (RFLP) assay and a PCR heteroduplex assay allowed the detection of grapefruit and mandarin juice in orange juice. Specifically, the PCR heteroduplex assay showed a better limit of detection of 2.5%, while a limit of detection of 10% resulted from using the RFLP assay [50]. Additionally, the detection of adulterated orange juice with mandarin juice performed by Aldeguer et al. [51], using a single nucleotide polymorphism (SNP) at the trnL–trnF intergenic region of the chloroplast chromosome as marker. As a result, a limit of detection of 5% added mandarin was achieved in both fresh and reconstituted orange juices [51]. A similar study was conducted by Pardo and Miguel Angel [52] who successfully determined the addition of mandarin in orange juice with the detection limit of 1%.

In the case of wine adulteration, DNA analysis is the most applied approach for identifying the varietal origin of grapes for wine production. In grapevine varietal identification, OIV has approved the use of simple sequence repeat (SSR) as nuclear molecular markers. However, SSR markers have revealed some limitation in terms of DNA quality relating to the possible inhibition of PCR reactions by large amount of polyphenols, polysaccharides, and proteins in must and wine and the degradation of DNA by alcoholic fermentation [53]. Recently, small molecular markers like SNP have been used to deal with DNA degradation. For instance, Boccacci et al. [54] used SNP genotyping assays to authenticate varietal origin of "Nebbiolo" musts and wines. Based on 1157 genotypes, two SNPs were sufficiently adopted for authentication. The developed assays allowed the authors to identify the must mixtures and wine mixtures at the sensitivity of 1% and 10–20%, respectively [54].

Overall, the advantages of DNA-based techniques are their sensitivity, quick analysis time, and convenience for large-scale measurement. Additionally, these approaches are also not affected by geography. However, the application of these techniques faces several problems, such as the degradation of DNA in fruit juices due to thermal processing under acid conditions and the removal of pulp in the clarification process of clarified juice leading to the difficulty in collecting DNA. Moreover, the heavy workload, the complex procedure for selecting molecular markers, and the high cost are other problems that must be taken into account [55].

**Table 3.** Typical application of DNA-based techniques for authentication of and adulteration detection in juice and wine.

| Sample | Aim | Accuracy | Reference |
|---|---|---|---|
| Grapefruit and orange juice | Detection of grapefruit juice in orange juice | Detection limit: 2.5–10% | [50] |
| Orange mandarin juice | Detection of orange adulteration with mandarin juice | Detection limit: 5% | [51] |
| Orange and mandarin juice | Determining addition of mandarin in orange juice | Detection limit: 1% | [52] |
| Musts and wines | Authentication of musts and wines | Detection limit: 10–20% for wine and 1% for must | [54] |

## 3. Non-Destructive Techniques

### 3.1. Spectroscopic Techniques

Spectroscopic techniques have advantages of being non-destructive and cost-saving and not requiring sample pretreatment. However, these techniques produce huge and redundant amounts of data, leading to a difficulty in data processing. Besides the information about the sample itself, the spectral data also contain irrelevant information, like information overlap, noise, baseline drift, etc., which will reduce the accuracy. To reduce the useless information, various methods of data pretreatment can be performed based on the situation, such as smoothing, derivative, baseline correction, standard normal variate (SNV), multiplicative scatter correction (MSC), and de-trend. Additionally, it is difficult for naked eye to discriminate between the pure sample and the adulterated sample due to the small difference in their spectral data. Therefore, chemometric methods are often applied to develop statistical models for analysis.

### 3.1.1. Infrared (IR) Spectroscopic Technique

Infrared (IR) signals are related to molecular vibrations. Infrared light irradiation results in the change of molecules' vibrational state. Each chemical substance produces a specific vibrational frequency in the IR region, which can be used as "fingerprint" to verify its presence in the sample. According to the spectral region, there are two main IR-based methods frequently applied, including near-infrared spectroscopy (NIRS) (14,000 to 4000 cm$^{-1}$) and mid-infrared spectroscopy (MIRS) (4000 to 400 cm$^{-1}$) [56]. Some applications of spectroscopic techniques for detecting adulteration of juice and wine are presented in Table 4. The applicability of NIRS for fruit juice and wine authenticity has been confirmed in previous studies. For example, in the case of juices, Calle et al. [57] achieved a detection limit of 5% when using a combination of NIRS and machine learning models to detect juice-to-juice adulteration of pineapple, apple, and orange juices with grape juice. Following that, an accuracy of up to 97.67% was obtained for their discrimination model (LDA and RF models), and their support vector regression (SVR) model achieved a prediction error of less than 1.7% for adulteration level [57]. Shafiee et al. [58] optimized data mining method on the NIR spectrum of lime juice to obtain the best classification result of 97% using support vector machine (SVM) method. In case of wine, Zaukuu et al. [59] used NIRS to develop discrimination models to differentiate original Tokaj wines and adulterated ones. Chemometric-processed data showed that the proposed method was able to determine grape must concentrate added to Tokaj wines with the lowest prediction error of 9.31 g/L for validation [59]. A recent study by Hencz et al. [60] has proven the feasibility of NIRS for the detection of wine adulteration in which they succeeded in detecting water dilution and addition of sugar in Portugieser and Sauvignon Blanc wines. Their method could detect adulterants with a prediction error of less than 0.504%, using partial least squares regression (PLSR) models [60]. However, no validation method was discussed in their work.

**Table 4.** Typical application of spectroscopic techniques for authentication of and adulteration detection in juice and wine.

| Sample | Technique | Aim | Accuracy | Reference |
|---|---|---|---|---|
| Pineapple, apple, and orange juices | NIRS | Detection of grape juice as adulterant in pineapple, apple, and orange juices | Detection limit: 5% | [57] |
| Lime juice | NIRS | Discrimination of natural and synthetic lime juice | Classification correction: 97% | [58] |
| Tokaj wines | NIRS | Differentiation of wines | Classification correction: 100% | [59] |
| Portugieser and Sauvignon Blanc wines | NIRS | Detection of wine adulteration with water and sugar | Detection limit of water: 28.57% Detection limit of sugar: 3.62% | [60] |
| Lime juice | NIRS | Differentiation of adulterated lime juices with citric acid | Classification correction: 88% | [61] |
| Grape juice | FT-IR | Detection of juice adulteration | Detection limit: 50–100% | [62] |
| Orange juice | FT-IR | Detection of water addition | Detection limit: 0.5–20.0% | [63] |
| Commercial juices | FT-IR | Detection of juice adulterated with saccharin | Detection limit: 0.1–2% | [64] |
| Red wine | FT-IR | Differentiation of wine | Classification correction: 62.96% | [65] |
| Cabernet Sauvignon wines | NIRS and MIRS | Differentiation of wine | Classification correction: 77–97% | [66] |
| Pineapple, orange, and apple juices | FT-IR | Detection of grapefruit as adulterant in pineapple, apple, and orange juices | Detection limit: 5% | [67] |
| White wine | FT-Raman | Differentiation of wine | Classification correction: 94.1–100% | [68] |
| Wine | FT-Raman | Wine authentication | Classification correction: 84–100% | [69] |
| Orange juice | NMRS | Detection of juice adulteration | Detection limit: 10% | [70] |
| Apple, orange, pineapple, and pomegranate juices | NMRS | Detection of juice-to-juice adulteration | Detection limit: 6.25% | [71] |
| Wine | NMRS | Differentiation of wines | Classification correction: 89% | [72] |
| Chinese red and white wines | NMRS | Identification of grape varieties | Classification correction: 82–94% | [73] |
| Wine | NMRS | Classification of wine | Classification correction: 96% | [74] |

Recently, portable and handheld NIR devices have drawn increasing attention in the food industry as online quality monitoring tools; however, their application in quality control of fruit juice and wine remains very limited. A handheld NIR spectrophotometer was applied by Jahani et al. [61] who achieved classification correction of 88% in detecting adulterated lime juices using PLS-DA method with SNV transforming. Additionally, Ehsani et al. [75] obtained the accuracy of 100% in authenticating orange juices using handheld NIR spectrometers. Despite their convenience in measurement, handheld NIR devices sometimes showed lower sensitivity than benchtop ones, and more studies are needed to confirm their applicability [76].

Along with NIRS, MIRS has been widely used in the validation of food authenticity. MIRS focused on the mid-infrared region. MIRS measures fundamental vibrations, while NIRS measures the overtones and combination bands; consequently, MIRS, corresponding to spectroscopic method referred as Fourier transform infrared spectroscopy (FT-IR), produces more information on the sample than NIR [77]. The feasibility of using FT-IR for the quantification of concord juice added to grape juice was demonstrated by Snyder et al. [62].

Accordingly, a prediction error of 5.55–8.40% was achieved using PLSR of FT-IR data [62]. Ellis et al. [63] reported the prediction error of 1.7% when they combined FT-IR with PLSR to quantify sugar concentration (0.5–20.0%) added to pure orange juice. The addition of saccharin into commercial fruit juices was investigated by Mabood et al. [64] who combined FT-NIRS and PLSR. The result of cross validation based on 30% of the total adulterated juice samples obtained root mean square error (RMSE) of 0.92% and correlation coefficient of 0.97 [64]. FT-IR spectroscopy was feasible for the classification of wines according to the year of vintage with a fairly good accuracy of 62.96% using the LDA model [65]. In another study, soft independent modelling of class analogy (SIMCA) model showed better performance than DA model in combination with spectral data acquired from MIRS and NIRS to classify wines from different countries. Accordingly, SIMCA model produced classification correctness of 92–97%, whereas DA model obtained accuracy of 77–86% [66]. Recently, Calle et al. [67] successfully applied FT-IR to disclose juice-to-juice adulteration, which is one of the most difficult frauds to detect. Consequently, their method reached a detection limit of 5% with discrimination accuracy above 97% and prediction error lower than 1.4% for validation [67].

### 3.1.2. Raman Spectroscopic Technique

Quality food assessment by this technique is based on measurement of the Raman scattering effect that is caused by the difference between the frequencies of incident and scattered radiation. The information acquired by measuring the inelastic scattered light emitted by molecular vibrations can be used as a fingerprint to determine different chemical substances in food samples. The main difference between Raman and infrared spectroscopy is that Raman spectroscopy acquires spectral data based on the vibration of molecules of scattering light, while IR spectroscopy obtains spectral data through light absorption. Additionally, Raman activity is the result of changing polarizability of a molecule, while IR activity is caused by the change of dipole moment of a molecule [78].

For quality control of wine, Raman scattering effect was investigated for different purposes, such as monitoring the fermentation process [79], determining wine compounds [80–82], and discriminating wines [68,83]. Mandrile et al. [83] used Raman spectroscopy for wine traceability according to grape varieties, geographical origin, and time of ageing. Their finding revealed that DA models obtained a discrimination rate of 90% for grape varieties and production area and 84% for time of ageing [83]. A recent study by Magdas et al. [69] used FT-Raman spectra for wine authentication to perform the discrimination of wines according to geographical origin, variety, and vintage. Accordingly, the correct classification rates were 84% for variety, 100% for geographical origin, and 90.7% for vintage [69]. Studies on applications of Raman spectroscopy for detecting adulteration of juices are rare.

### 3.2. Nuclear Magnetic Resonance (NMR) Spectroscopic Technique

NMR spectroscopy has been considered as one of the most commonly applied analytical techniques for the evaluation of food quality. NMR spectroscopy is based on the measurement of the energy absorption of atomic nuclei with non-zero spins under the effect of a magnetic field. The effect of the nuclei of surrounding molecules on the energy absorptions of the atomic nuclei results in small local changes to the external magnetic field. Thus, NMR spectroscopy is able to produce detailed structural information of the molecules in food samples since there is a relation between the observed interactions of an individual atomic nucleus and the atoms surrounding it [56]. Among the NMR techniques, $^1$H measurement over $^{13}$C is likely to be the most frequently used technique in the quality assessment of beverages thanks to its high sensitivity and short relaxation times [84].

NMR spectroscopy has been widely used to identify juice and wine adulteration for decades. For instance, Spraul et al. [85] used a 400 MHz flow-injection NMR spectrometer to differentiate juices. This method allowed them to determine 28 different chemical compounds, and a relative accuracy of nearly 10% for more than 95% of the samples

was confirmed [85]. Vigneau and Thomas [70] demonstrated that $^1$H-NMR spectroscopic method combined with PLSR was able to achieve the prediction error of 3.47% when they attempted to discriminate pure orange juice and the ones adulterated with clementine juice. Recently, Marchetti et al. [71] combined $^1$H NMR with PLSR to determine the proportions of apple, orange, pineapple, and pomegranate juices in their blends. PLSR model achieved the best performance with the prediction error of less than 10% and $R^2$ of 0.821–0.987 [71].

In the case of wine, authentication can be carried out based on the information acquired from the spectra of $^1$H, $^2$H, and $^{13}$C. NMR spectroscopy has been used for wine authenticity determination in several studies for over a decade. For instance, Godelmann et al. [72] applied $^1$H NMR spectroscopy together with multivariate data analysis to differentiate German wines originating from different grape varieties and geographical origins and of different ages. The results showed that the correctness of classification varied from 89% to 97% [72]. Another example is the study by Fan et al. [73] that demonstrated $^1$H NMR spectroscopy to be an effective tool for authenticating wines. The use of LDA method on spectral data resulted in average correct classification rates of 82–94% for red and white wines [73]. Recently, a study of wine classification according to color and content of residual sugar was performed by Mascellani et al. [74], who used machine learning methods to build classifier models based on acquired $^1$H NMR spectra. As a result, the models achieved the classification correction rate of 93% [74].

### 3.3. Electronic Techniques

Recently, electronic sensors imitating human senses have been used extensively for food quality evaluation, especially electronic nose (e-nose) and electronic tongue (e-tongue). While e-nose uses an array of sensors to identify and differentiate odors in complex food matrices, e-tongue uses a set of chemical sensors to detect and classify chemical substances in liquid samples. After acquiring the signals, multivariate data analysis is applied to build statistical models for further qualification and quantification [86]. Some typical examples of studies applying e-tongue and e-nose on the detection of juice and wine adulteration are presented below. Table 5 presents some applications of e-nose and e-tongue for authenticating and detecting adulteration in juice and wine.

**Table 5.** Typical application of electronic techniques for authentication of and adulteration detection in juice and wine.

| Sample | Technique | Aim | Accuracy | Reference |
|---|---|---|---|---|
| Tomato concentrate | E-tongue | Detection of tomato concentrate adulteration | Detection limit: 0.5% | [18] |
| Tokaj wine | E-tongue | Differentiation of wine | Classification correction: 100% | [59] |
| Lime juice | E-tongue | Detection of adulteration | Detection limit: 5% | [87] |
| Apulian red wines | E-tongue | Differentiation of wine | Classification correction: 70% | [88] |
| Cherry tomato juice | E-nose | Differentiation of juice | Classification correction: 79.53% | [89] |
| Apple, lemon, and sour cherry juices | E-nose | Differentiation of juices adulterated with alcohol | Classification correction: 95% (LDA) Classification correction: 98.33% (SVM) | [90] |
| Orange juice | E-nose | Detection of freshly squeezed orange juices adulterated with concentrated orange juices | Detection limit: 0–30% | [91] |
| Cherry tomato juices | A combination of e-tongue and e-nose | Detection of adulteration | Detection limit: 10% | [92] |
| Spanish wine | E-nose | Differentiation of wine | PCA Classification: 91.3% using first two principal components (PCs) | [93] |

### 3.3.1. E-Tongue

Since tomato concentrate is an important ingredient in food processing, it is usually adulterated by water dilution and addition of cheaper bulking agents and other food additives. Vitalis et al. [18] applied e-tongue to detect adulterants in concentration of 0.5–5%. The used e-tongue showed good ability to detect adulterants with LDA classification correctness above 75.72%. PLSR models for quantification achieved excellent quality of results with prediction error of less than 1% [18]. In another study, Bahrami et al. [87] applied e-tongue combined with different statistical models to detect adulteration of lime juice. Their results indicated that multilayer perceptron (MLP) model had better performance than SVM model with a correctness of up to 99.33% and prediction error of less than 0.1% in estimation of adulteration levels [87].

With regard to wine authentication, Lvova et al. [88] achieved a correct discrimination of above 70% when combining a potentiometric e-tongue system with PLS-DA model for evaluating the brand uniformity of Apulian red wines. However, their small sample could not ensure the adequate robustness of the classification model. In another study, e-tongue was used by Zaukuu et al. [59] to differentiate high-quality Tokaj wines from lower-quality ones that were altered with grape must concentrate to satisfy the requirement of sugar content. The correct classification rate of 100% for adulterated and unadulterated wines was achieved using e-tongue combined with LDA method [59].

### 3.3.2. E-Nose

In the last decade, the number of studies on the application of e-nose in adulteration detection and authentication of fruit juice and wine are quite limited. A typical study on the application of e-nose for detecting adulteration of tomato juices was performed by Hong et al. [89] who performed various clustering methods on spectral data. They concluded that the spectral clustering showed statistical significance, and the accuracy of adulteration detection reached 79.53% $\pm$ 3.56% [89]. In addition, Ordukaya and Karlik [90] successfully classified various juice samples adulterated with alcohol using a cyranose e-nose setup with 32 polymer sensors. The effectiveness of two classifier models (LDA and SVM) were compared. The result of classification showed that SVM (98.33%) performed better than LDA (95%) [90]. Another application of e-nose in adulteration detection of juice is reported in the study by Shen et al. [91]. A combination of e-nose with PCA and LDA model was used to discriminate freshly squeezed orange juices and those adulterated with orange juice concentrate in concentration of 10–30%. The LDA model achieved an overall accuracy of 97.9% in calibration set and 91.7% in validation set [91]. However, there was a concern about the generalization of the LDA model since their sample sizes for calibration and validation were quite small.

When dealing with complex samples, exclusive usage of electronic nose (e-nose) or electronic tongue (e-tongue) data is inadequate. Therefore, multisensor data fusion techniques with combination of e-nose with e-tongue were tested. Hong et al. [92] developed a multisensor data fusion method that combined e-nose and e-tongue to identify and quantify adulterants in cherry tomato juices with the smallest detectable level of 10%. The results of their work showed that the application of both instruments could improve the performance as long as the appropriate data fusion methods are used. The prediction error for soluble solid content in validation for both principal component regression and multiple linear regression was below 0.09%. Their finding also highlighted the importance of standardization method of acquired data [92].

The potential application of e-nose for wine discrimination was confirmed in a study by Bellincontro et al. [93]. The e-nose based on an array of eight quartz microbalances was employed to differentiate five wine types produced from the same variety of grape with various ethanol concentration (11.5 to 18%) and different contents of volatile compounds. The unsupervised method of PCA was used to evaluate the discrimination ability of e-nose on wine types. Four in five groups of wine were successfully distinguished using the first two PCs (PC1: 83.8% and PC2: 7.50%) [93]. Since the used PCA model is just suitable

for screening the recognition ability, the main limitation of their work is that supervised classification models were not used for the evaluation of discrimination. Additionally, the lack of validation of their work may lead to the unreliable confirmation of the applicability of the proposed technique.

The advantages of e-senses are the sensitivity, the low cost, and the minimal requirement of sample preparation. In spite of that, a disadvantage of these techniques is that their accuracy is affected by the environmental factors, including temperature (for both e-nose and e-tongue) and humidity for e-nose, which are able to cause the sensor drift [94]. Moreover, these techniques require more complicated procedures to operate than other non-destructive techniques like NIRS [18].

### 3.4. Imaging-Based Techniques

Imaging-based techniques have proven their effectiveness in quality assessment of food for decades; however, the application of these techniques for the detection of food fraud is quite rare. Among the imaging-based techniques, digital image analysis (DIA) and light backscattering imaging (LBI) are the two techniques that have been utilized for the determination of adulterants in wine and juice. Table 6 summarizes examples of application of imaging-based techniques for authentication of and adulteration detection in juice and wine.

**Table 6.** Typical application of imaging-based techniques for authentication of and adulteration detection in juice and wine.

| Sample | Technique | Aim | Accuracy | Reference |
|---|---|---|---|---|
| Gran Reserva wine | DIA | Detection of adulterated wine | Detection limit: 2.3% | [95] |
| Physalis juice | DIA | Detection of juice adulteration | Detection limit: 20% | [96] |
| Orange and mandarin juices | DIA | Differentiation of juices | Classification correction: 83–97% | [97] |
| Red and white wines | LBI | Detection of wine adulteration by water dilution and the addition of sugar | Classification correction: 53.33–76.67% for water addition and ≥93.33% for sugar addition. | [60] |

### 3.4.1. Digital Image Analysis

Quantifying visual food parameters and connecting them with the quality are promising methods in food research, because visual properties are one of the most important quality factors. Digital image analysis is the technique that transforms visual information of color, shape, and pattern to numeric parameters. The combination of color data extracted from a set of digital images and color spaces, such as RGB (red, green, blue) and HSV (hue, saturation, value) can create a data matrix. Then, applying chemometric tools will process the numerical information from color space to qualify and quantify the food quality [98]. Multivariate Image Regression (MIR) and Multivariate Image Analysis (MIA) are the two most common computational methods for processing color data [95]. The effectiveness of digital image analysis was confirmed by Licodiedoff et al. [96], who developed linear models using the image analysis based on RGB system to investigate the dilution of Physalis juice with concentrations ranging from 0 to 100%. Applying the same approach, Stinco Scanarotti et al. [97] evaluated the differences in color of orange and mandarin juices from several varieties. The authors concluded that the method allowed the differentiation between juices with 83–97% correct classification [97]. Moreover, the feasibility of digital image analysis on identifying dilution levels of orange juice was studied based on the color specifications, in which the thickness of sample was investigated and kept constant to make color measurements reproducible and reliable. The results revealed that there was significant correlation between digital image analysis results provided by the instrument and visual analysis by panelists ($p < 0.05$), and orange juice with different dilution levels could be detected correctly [99]. Similarly, the applicability of RGB digital images on

qualification and quantification of adulterations in aged wine was confirmed in a study by Herrero-Latorre et al. [95]. The wine samples included pure wines (Gran Reserva, Crianza, and Joven) and synthetic adulterated Gran Reserva. Multivariate image analysis was able to recognize differences among the wines and predict adulteration level with the detection limit of 2.3%. The drawback of digital image analysis is that it is not suitable for detecting adulteration based on internal quality attributes since it lacks spectral information. A combination with other techniques like spectroscopy would improve its performance.

### 3.4.2. Light Backscattering Imaging

Among imaging techniques, novel techniques of light backscattering imaging have recently proven their applicability in monitoring the quality of various agricultural commodities. The technique is classified into three categories, including laser light backscattering imaging (LLBI), multispectral backscattering imaging (MBI), and hyperspectral backscattering imaging (HBI), according to the image acquisition system and the wavelength range [100]. When the light penetrates food matrices, it can interact with the internal components by absorption and scattering or transmission to the other side of the surface. While the light absorption relates to chemical compounds of the food such as pigments, water, sugar, etc., the scattering results from the collision of photons inside media [101]. Although the technique has been widely investigated for quality evaluation of food and agricultural products, its application for the detection of beverage adulteration draws little attention from researchers. The very first study on the applicability of LLBI for the assessment of wine adulteration was performed by Hencz et al. [60]. Adulteration simulation was performed by the addition of water, sugar, and both. The results revealed that laser backscattering signal responded sensitively to the adulteration based on ANOVA F-score. The proposed technique showed better performance in the detection of sugar addition compared with water dilution. The classification of adulterated wine samples was performed using LDA. At all wavelengths, the classification accuracy varied in the range of 53.33–76.67% for water addition and 93.33–100% for sugar addition. The prediction ability of generalized linear model regression indicated lower prediction error for sugar adulterated wines at 3.06% than those for water diluted wines at 20.39%. The authors have demonstrated the feasibility of LLBI for wine authentication; however, their work revealed some limitations such as using small sample size and lacking appropriate validation of calibration models.

According to the best of our knowledge, there have been no studies aiming at the application of light backscattering imaging for the detection of fruit juice adulteration.

## 4. Conclusions and Outlooks

Fruit juice and wine have become one of the most popular targets of food fraud and mislabeling. Different analytical techniques have been developed to detect the adulterants in juice and wine. According to the reviewed literature, the proposed methodologies were classified as destructive and non-destructive techniques. The most common destructive analysis techniques include physicochemical methods, isotope analysis, elemental analysis, chromatographic techniques, and DNA-based techniques. Although they have relatively high sensitivity and accuracy, the main disadvantage of these approaches is the requirement for sample destruction and pretreatment in the process of operation. As a result, the analytical procedure is time-consuming, complex, and incompatible for real-time quality control. Furthermore, the instability of some marker compounds, like phenolic compounds, DNA, and isotope ratios during processing and storage, may produce variations in analytical results.

To overcome the limitation of destructive analysis techniques, various non-destructive techniques have been developed by researchers so far, including spectroscopic techniques, electronic techniques, and imaging-based techniques. These techniques are based on the investigation of the general chemical profile of studied samples. After acquiring the relevant signal, multivariate analysis methods are necessary to process the data and

generate qualitative or quantitative models. Compared with destructive techniques, non-destructive techniques have the advantages of little or no sample preparation, ease of use, reduced cost, high portability, and no need for reagents. Especially, some spectroscopic techniques have very rapid analysis that is performed within 1 min [57]. Although the feasibility of these techniques in adulteration detection and authentication was proven by many authors, there were challenges for applying these methods in real life. Indeed, most of the studies were performed under laboratory conditions with small number of samples. Moreover, the robustness of calibration models greatly depends on the selected methods of signal pretreatment, feature selection, and modeling. To achieve universal models for industrial application, more research needs to be conducted focusing on increasing the number of samples for calibration and validation and selecting optimal multivariate analysis methods.

**Author Contributions:** Conceptualization, H.X.M. and L.L.P.N.; formal analysis, H.X.M., N.T.T.H. and T.T.P.; resources, L.L.P.N., L.B. and L.F.; data curation, H.X.M.; writing—original draft preparation, H.X.M., N.T.T.H. and T.T.P.; writing—review and editing, L.L.P.N. and L.B.; visualization, H.X.M., N.T.T.H. and T.T.P.; supervision, L.L.P.N. and L.B.; funding acquisition, L.F. All authors have read and agreed to the published version of the manuscript.

**Funding:** This research received no external funding.

**Data Availability Statement:** No new data were created or analyzed in this study. Data sharing is not applicable to this article.

**Acknowledgments:** The authors acknowledge the Doctoral School of Food Science of the Hungarian University of Agriculture and Life Sciences for the support provided for this study.

**Conflicts of Interest:** The authors declare no conflict of interest.

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
