# Peer review of "Current Techniques for Fruit Juice and Wine Adulterant Detection and Authentication"

_beverages, doi:10.3390/beverages9040084_

Round 1

Reviewer 1 Report

I recommend major changes in the manuscript in terms of structure and the number, relevance and updating of references. Some of these changes are:

LINE 20 add also that wine is adulterated.

LINE 58 define FDA.

LINE 88 define AIJN, furthermore this organism is named without defining it in line 33.

LINE 89 add reference.

LINE 91 the abbreviation TSS has been previously defined on line 86.

LINE 93 and 98 can be used directly with the abbreviation TSS.

LINE 190 add the references of the studies

Review all abbreviations, e.g., Line 191 and 196 same defined abbreviation (HS-SPME).

LINE 212 use the abbreviation.

LINE 215 is redefined HPLC.

LINE 229 is redefined HPLC.

LINE 240 is redefined HPLC.

LINE 271 is redefined RFLP.

LINE 283 is redefined OIV.

LINE 288 SNPs is defined although it appears previously in line 272-273

Section 3.1.1 I recommend adding some current and relevant references to the subject that are also more current, such as:

“Rapid Detection and Quantification of Adulterants in Fruit Juices Using Machine Learning Tools and Spectroscopy Data”

“Detection of Adulterations in Fruit Juices Using Machine Learning Methods over FT-IR Spectroscopic Data”

Do a major review of the abbreviations and the readability of the text. I recommend restructuring some parts and avoid using abbreviations such as "PJC" and "GJC" that have no relevance and appear only in 4 lines of text.

LINE 336 is defined PLSR when it has been used previously.

LINE 387 PLS is defined again.

LINE 389 coefficient of determination define it before.

LINE 399 is defined LDA but has been previously used several times.

LINES 404-416 I would introduce them in another sentence or at the beginning of the section since they are not "conclusions" of the section as such.

LINE 432, PLS is defined again.

TABLE 5. Add further and more current studies.

I recommend restructuring the conclusions. Note that non-destructive techniques need not be simpler in terms of sample preparation or faster than destructive techniques. This is stated at some point in the text.

Author Response

Please find answers to all reviewers in the attached file.

Reviewer 2 Report

Paragraph 88: Abbreviation used without explanation

Only 3 publication for physiochemical methods, more input. 

 2.2 Isotope analysis: try revising this chapter it needs more input. Isotope fingerprints is offering an analytical technique that provides results with high precision and accuracy  for the investigation of wine adulteration.    

Paragraph 112: you are citing a statement from a review, find an original paper that has published research work regarding your statement. 

 Table 1 to short, add more publications from your publication list.

 Omit reference 19, or you like to publish "negative data". 

 2.3 more input 

 Paragraph 149: not all elements analyzed  are representative for distinguishing the origin of a wine. Find out what do authors recommend which ones (five of them ) are to be considered. 

Elaborate in the same manner the next cited reference no 23.  

 2.4 Introduction to the Chromatographic: Consider paragraph revising 

Paragraph 172: consider English editing 

Liquid Chromatography: Concluding paragraph

NMR and other spectroscopic analysis are  added lately in addition to complete the phenolic imprint in wines  

Paragraph 376: reference missing

Paragraph 404: I would not completely agree with this paragraph. NMR is never considered one of the easy and quick in operation. Just one 1H SNIF-NMR measurement last 4 hours.

Line 406: Typing error: NMR instead of NIR. 

 General recommendation: This is a review article and should review at least what is done in the field of research in the last decade. There are more than 20 papers older than 10 years period. 

I think the English is good

Author Response

(The authors gave the same response as above.)

Reviewer 3 Report

The work of dr. Hoa Xuan Mac et al entitled' Current techniques for adulterant detection and authentication of fruit juice and wine'  nicely presents the principle of adulteration detection, the application and performance and the advantages and limitations of various analytical techniques. The most common destructive analysis techniques described  include physicochemical methods, isotope analysis, elemental analysis, elemental analysis, and DNA-based Techniques and the authors conclude that these are time-consuming, complex and incompatible for real-time quality control. On the other hand, nondestructive techniques have the advantages of simple sample preparation, rapid detection and high efficiency but more studies are required to improve the performances of these techniques. Consequently, there is still room for developments of quick, nondestructive analytical methods.

We recommend that the authors include a subsection describing the juices and the wines they refer to further in their manuscript.

Author Response

(The authors gave the same response as above.)

Round 2

Reviewer 1 Report

I recommend the publication of the manuscript in its present form.

Reviewer 2 Report

thanks for adressing our issues!!